# OpenReview forum: "Teach a Reward Model to Correct Itself: Reward Guided Adversarial Failure Discovery for Robust Reward Modeling"
_ICLR.cc/2026/Conference — ICLR 2026 Conference Withdrawn Submission_

### Official Review · Reviewer_UWhQ · 2025-10-31

**Soundness:** 2
**Presentation:** 2
**Contribution:** 2
**Rating:** 2
**Confidence:** 3

**Summary:**

Reward Model trained with human preferences are important for aligning LLMs but can break with distribution shift. The paper introduces a preference distribution agnostic approach to use the reward model to guide control decoding toward mis-specified responses.  Using this approach, they train on HH-RLHF and Beaver-tails and shows improvements of 35-45% across PPO, DPO and Best of N (rejection sampling).

**Strengths:**

1.	Targeted correction is reasonable since a small proportion of the samples might be mislabelled.
2.	The approach of constrained decoding to generate synthetic data is reasonable to train models to avoid tokens that have high likelihood of harm (e.g. vulgar words or words commonly associated with crime).
3.	There is a rich diversity of illustrations in the main paper and appendix to clarify the work. I would recommend putting a few illustrated examples (ideally at the front) to better situate the work.

**Weaknesses:**

1.	The approach described in Fig. 2 (and Eq. in line 64) suggests that there’s a clean line separating chosen and rejected responses (into 2 distinct classes). However, preferences are only ordinal (and not cardinal) – meaning to say that we can have a great chosen vs good rejected - or - good chosen vs bad rejected. This means that class-conditioning doesn’t necessary work because in some settings where we have 4 responses A>B>C>D, response B will fail in both chosen and rejected baskets. (I recognize that this issue is more applicable to helpfulness than harmlessness,  but might still happen with harmful settings where one is more harmful than another).
2.	The assumption that we can use a reward model as a value function by only giving it a partial completion up to a certain token is poorly supported. The paper cites Khav et al. (2024) – which shows reward of partial responses can be used to guide decoding. However, in my experience training and inferencing with reward models, I think it’s very unlikely for Reward Models to accurately grade partial responses (since they were never trained to do so). Can the authors provide further evidence that this approach does indeed work? (I recognize that this might be less of a problem for harmlessness but would still like some evidence of this).
3.	The paper doesn’t use standard safety benchmarks such as ToxiGen [1] or XSTest [2] to evaluate the performance of their aligned models. Instead, it only measures it based on the predicted rewards of the reward models, which could be overfitted to. This makes it hard to compare this work with other approaches.

[1] Thomas Hartvigsen, Saadia Gabriel, Hamid Palangi, Maarten Sap, Dipankar Ray, Ece Kamar. ToxiGen: A Large-Scale Machine-Generated Dataset for Adversarial and Implicit Hate Speech Detection ACL 2022

[2] Paul Röttger, Hannah Rose Kirk, Bertie Vidgen, Giuseppe Attanasio, Federico Bianchi, and Dirk Hovy. Xstest: A test suite for identifying exaggerated safety behaviours in large language models. NAACL 2024.

**Questions:**

1.	How does the approach compare to methods such as Rule-based Rewards[1] or DExperts[2]?
2.	Not exactly a question but I recommend that the authors mention that the paper is on safety-based Reward Modelling early in the paper (ideally in the abstract) – this wasn’t clear to me until I got to line 268. Prior to that, my impression was that the paper was applicable to both helpfulness and harmlessness (which inspired weakness 1 and 2).
3.	Not exactly a question but some of the figures (5 to 9) are less informative (not clear what information they are seeking to communicate) and could be better presented in a table instead.

[1] Tong Mu, Alec Helyar, Johannes Heidecke, Joshua Achiam, Andrea Vallone, Ian Kivlichan, Molly Lin, Alex Beutel, John Schulman, Lilian Weng. Rule Based Rewards for Language Model Safety. NeurIPS 2024.

[2] Alisa Liu, Maarten Sap, Ximing Lu, Swabha Swayamdipta, Chandra Bhagavatula, Noah A. Smith, Yejin Choi. DExperts: Decoding-Time Controlled Text Generation with Experts and Anti-Experts. ACL 2021.

---

> ### Author Response · Authors · 2025-11-15
> **Author rebuttal to Reviewer UWhQ (1/n)**
>
> * **Weakness 1**
>     * **Figure clarification** - Figure 2, was included purely for illustrative purposes, depicting the concept of reward correction. The class can be interpreted as a threshold defined by the axioms that make up the true preference distribution and the dividing line is on how well the reward rates these samples accordingly. We will clarify this in the figure caption.
>     * **Dataset selction** - In our setup, class conditioning is determined by the prompt together with its associated preference distribution. This distribution may contain multiple objectives and may exhibit inherently variable criteria depending on user-specific preferences. We use the harmlessness dataset only as an illustrative example, because harmlessness can be quantified through well-defined axioms, making evaluation feasible. In this dataset, preference pairs are constructed based on which response is judged to be less harmful. As shown in Figure 4, our method produces samples that exhibit class consistency with respect to this objective.
>
>     * **Why REFROM is a necessary solution** - Identifying class-consistent samples is challenging when class definitions become arbitrary due to the absence of an explicitly defined preference distribution. To address this, we propose using a preference-aligned and preference-misaligned policy as a mechanism for generating class-specific samples, serving as practical proxies for the underlying preference classes.
>
>     * **No need for explicity preference knowledge** - In this setup, we treat the provided preferences as ground-truth labels and do not attempt to interpret the underlying preference distribution, which may vary significantly across tasks or users. Our goal, instead, is to generate samples that would be aligned with the distribution regardless of the particular axioms or criteria on which it was constructed. The framework is therefore designed to produce preference-consistent responses without requiring any explicit understanding or modeling of the distribution itself.
>     * **Practicality** - That said, in practical deployments, one could still re-label the generated samples through an additional round of human feedback collection. In this scenario, the framework serves as a tool to narrow down the subset of samples that require re-labeling, thereby reducing the overall annotation cost.
>
> <!-- For evaluation, however, explicit knowledge of the preference distribution is necessary. This is precisely why we employ an RLHF dataset corresponding to harmlessness, where class consistency can be evaluated according to predefined axioms. As shown in **Figure 4**, samples generated under our proposed method adhere to the definitions of harmfulness and harmlessness when evaluated using Gemini. -->
> ---
>
> * **Weakness 2**
>
>     * **Goal** - The goal of our work is to demonstrate that within the framework of maximum likelihood–based reward modeling (specifically, the Bradley–Terry formulation), a trained reward model can exhibit vulnerabilities to samples lying outside its training distribution.
>
>     *  **Contribution** - We argue that these out-of-distribution samples can be identified through the reward model itself and subsequently corrected. ***We do not calim this to be a method to universally find all possible out of distribution samples*** as it would be intractble.Thus our objective is not to present a frontier-level reward model as we are neither adapating an existing model nor training with a foundational reward dataset, but rather to reveal and mitigate such vulnerabilities inherent in the reward modeling process. This is the reason why we do not evaluate on samples that are vastly out of distribution and instead evaluate on ***a held-out test set*** which were never seen during the training (thus removing the concern of overfitting), where we show that our approach effectively reduces the vulnerabilities that a naive reward modeling objective is prone to.
>
>
> The novelty of our work thus stems from two main contributions:
>
>
> 1. ***Automated identification of reward failures***: We show that reward failure modes can be detected in an automated and self-contained manner, without requiring external heuristics or manually crafted prompting strategies commonly used in prior works.
>
> 2. ***Data-driven correction of reward vulnerabilities***: We demonstrate that the data points generated through our framework can be effectively used to improve the reward model, yielding better robustness and alignment compared to a naïve reward modeling objective.
>
>
> **On the suggested benchmarks** - The benchmarks of ToxicGen and XsTest are benchmarked designed towards a generational model as opposed our case were we consider the process of reward modelling rather. Past works [1] in the litreture of reward model failure have evaluated the reward models based on the predicted reward values with the rejected or chosen response as a baseline.
>
> ---

---

> > ### Author Response · Authors · 2025-11-15
> > **Author rebuttal to Reviewer UWhQ (2/n)**
> >
> > * **Weakness 3**
> >
> > We provide sample code for failure mode generation as evidence for the viability of our approach:
> > Code: https://anonymous.4open.science/r/sample_code/README.md
> >
> > Additional qualitative examples appear in the Textual Examples section of the appendix.
> >
> > * **Why is it vaible** - Our optimization is restricted to the top-k tokens (k = 5) from the aligned or misaligned policy, rather than the full token space (e.g., 128K tokens). This greatly increases the likelihood that the generated responses remain within the underlying distribution. Moreover, we regularize the optimization using the log-likelihood of the candidate continuation (see Equations 6 and 7), which preserves the coherence.
> >
> > * **Why trajectory reward** - We use a trajectory-level (outcome-based) reward model rather than a token-level reward model to maintain generality and avoid assumptions tied to a particular decoding scheme. Moreover, the method can be extended to other forms of controlled decoding, including integrating a learned value function or evaluating the reward on partially completed responses at each decoding step.
> >
> > * The method ***can also be extended to other from of controlled decoding*** including training a value function/ evaluating the reward on completed responses at each decoding stage.
> >
> > ---
> >
> > * **Question 1**
> >
> > Rule-based reward models explicitly require strong prior knowledge of the underlying preference distribution. Consider a user-specific LLM correction tasks: the user’s preferences are inherently arbitrary and difficult to formalize as a fixed set of rules. In such settings, constructing rule-based criteria that fully captures the preference distribution is infeasible. In contrast, we propose a general framework for reward modeling that can operate on any preference distribution, without requiring manually specified rules.
> >
> > Furthermore, methods such as DEXPERTS focus solely on controlled decoding and do not provide a mechanism for discovering failure modes (FM) of a reward model as these  FM are both model and data dependant.
> >
> > ---
> > * **Question 2**
> >
> > We adopt the safety modeling setup solely because it provides a practical evaluation environment, enabled by the existence of well defined safety axioms. However, our framework is fully general and can be applied to reward modeling under any preference objective.
> >
> >
> > ---
> >
> > We thank the reviewer for the suggestion with regards to the structuring of the paper and we will modify the manucript accordingly. In light of these considerations, we kindly request the reviewer to re-evaluate the significance of our contribution as a general framework towards identifying failure modes in reward models.
> >
> > [1] Junqi Jiang, Tom Bewley, Saumitra Mishra, Freddy Lecue, and Manuela Veloso. Interpreting language
> > reward models via contrastive explanations, ICLR 2025

---

> > > ### Comment · Reviewer_UWhQ · 2025-11-24
> > > **Reply to author response**
> > >
> > > Post-author-response, I'm even more confused about the claimed contribution(s) of the paper. The paper states `We use the harmlessness dataset only as an illustrative example, because harmlessness can be quantified through well-defined axioms, making evaluation feasible.` and also `In light of these considerations, we kindly request the reviewer to re-evaluate the significance of our contribution as a general framework towards identifying failure modes in reward models`.
> > >
> > > What this says is that the authors can only do experiments on harmlessness but they claim that it's a general framework - this is an example of over-claiming something that cannot be tested experimentally. As a researcher who works on helpfulness reward modeling, I cannot accept/endorse this framework's usefulness to helpfulness reward-modeling without experimental results. Furthermore, I believe that helpfulness also has many empirical ways/axioms to test (see for instance RewardBench [1] or RM-Bench [2]. My advice is to substantially re-frame the paper to focus on only safety reward modeling because this paper only has experiments on OR do some reward modeling experiments on helpfulness datasets too.
> > >
> > > It's not clear to me that the authors are making a concerted efforts to address my initial concerns. Responses to weakness 1-3 are not exactly answering my questions (Weakness 3 also has a dead link `https://anonymous.4open.science/r/sample_code/README.md` even though I didn't ask for the code; the end of Weakness 1 has `[object Object]`).
> > >
> > > Responses to Questions 1 and 2 are related to my original questions but I'm not convinced by the explanation either as I was anticipating additional empirical results, which the authors did not provide. As a note, `Past works [1] in the litreture of reward model failure have evaluated the reward models based on the predicted reward values with the rejected or chosen response as a baseline.` is not adequate justification for skipping certain experiments.
> > >
> > > In this case, I will be keeping my initial assessment but I hope the authors can put in some efforts to improve in their communicating their contribution clearly and (more importantly) accurately.
> > >
> > > [1] RewardBench: Evaluating Reward Models for Language Modeling. NAACL Findings 2025.
> > >
> > > [2] RM-Bench: Benchmarking Reward Models of Language Models with Subtlety and Style. ICLR 2025.

---

### Official Review · Reviewer_7WHV · 2025-11-01

**Soundness:** 3
**Presentation:** 2
**Contribution:** 2
**Rating:** 2
**Confidence:** 3

**Summary:**

In this papers authors highlight the failures of exisiting Reward Modes (RMs) under distribution shifts or adversarial prompts. They propose a lightweight method called REFORM which lets the RM essentially find its own blind spots, by using a clever decoding approach to find reponses from policy where RM doesn’t flip the preference but the reward is very low. So this guided decoding finds “low-reward preferred or high-reward non-preferred responses”. These adversarial examples are then used to make the model more robust by finetuning on top 5% of the failures.

**Strengths:**

- Simple light-weight and low cost approach to find adversarial examples for RM by using RM’s own scoring to guide controlled decoding
- Well formulated objective with clear explanation of the components involved
- It is an automatic approach and doesn’t need any human labelling to find adversarial examples
- Thorough evaluation setup and experiments which highlight the benefits of using the robust RM from their approach. Especially experiments on downstream Alignment show no degradation.

**Weaknesses:**

- Novelty  is limited since [1] has a similar approach of adversarial example generation followed by reward-model augmentation to improve robustness.
- The authors approach is lightweight but potentially limiting.  Work like [1] which uses RL optimization over a discrepancy/loss like R_1 - R_2  can potentially  explore much more diverse attacks leading to much more robust-RM, ofc at the cost of much more computation overhead. Comparisons to this baseline could be helpful.
- Minor: Models used for evaluation are old and not sota open LLMs
- Writing is a bit convoluted and can be improved upon

[1] https://arxiv.org/abs/2504.06141v2 , Adversarial Training of Reward Models

**Questions:**

- Rather than a token-level decode is it possible to do a high-temperature sampling from the policy to elicit the same behaviour. And possibly those discovered failures reflect the sequence level reward hacking examples encounterd in RLHF training better.
- Its unclear if the RM reward for partial completions will correlated with the final reward for the full completion. Can authors clarify this point.

---

> ### Author Response · Authors · 2025-11-15
> **Rebuttal to Reviewer 7WHV**
>
> * **Weakness 1**
>
> Our work differs from the cited work which we would (refer as AdvReward) due it's following weaknesses
>
> 1. **Computational cost** - As noted by the reviewer, the AdvReward method introduces additional computational overhead compared to our approach..
>
> 3. **Requires an ensemble of rewards** - The AdvReward framework also requires access to a strong ensemble of reward models, which is often impractical—especially in settings where a developer has access only to a preference distribution and no pre-trained reward models for that objective. In such scenarios, it is unclear how AdvReward could be formulated or applied. In contrast, our method operates in a ***fully self-contained manner***. Across multiple preference distributions and across different reward model families, each trained from scratch using the Bradley–Terry objective, we show that our approach consistently identifies and corrects reward failures. Importantly, our method does not rely on a large or highly specialized pre trained reward model. Instead, it remains effective even in simpler and more realistic settings where only the preference data is available.
>
>
> 3. **Limited scope** - The objective of AdvReward is formulated specifically to find out-of-distribution samples that receive low scores under a target reward model. In contrast, our objective is more flexible: we can generate both ***low-scoring preferred responses*** and ***high-scoring non-preferred responses***, while still ensuring class consistency, as demonstrated in Figure 4
>
> **Novely**: The key novelty of our work in failure mode identification lies in showing, for the first time, ***that a single reward model trained on a given preference distribution is sufficient to generate out of distribution (OOD)*** samples for both preferred and non-preferred classes. Moreover, we demonstrate that these generated OOD samples can be directly used to correct the corresponding reward model. This establishes that failure modes can be identified and mitigated without requiring ensembles, external heuristics, or multiple reward models.
>
>
> ---
>
> * **Question 1 and Question 2**
>
> We frame our objective as a controlled decoding framework. We show that even in the basic case of constrained decoding—similar to ARGS [1], which relies on a trajectory level reward model to score partial responses, our method remains effective.
>
> That being said the methods can be extended to any other controlled decoding frameworks. Namely
>
>    * **Token level reward models** [2] - In settings where a reward model is trained to provide token-level reward attribution, such models can enable richer forms of controlled decoding. However, we focus on the trajectory-level reward formulation for the sake of generality, as trajectory-based reward models are more commonly used in practice.
>    * **Value function based controlled decoding** [3] - With additional computational cost, these methods often train a value function on top of the reward model to enable richer decoding strategies. Moreover, there exist even more expensive controlled-decoding approaches in which decoding decisions are made based on evaluating fully decoded sequences.
>
> In all these scenarios, the current method can be naturally extended to operate over the top $k$ candidate tokens during generation. (There references are mentioned on detail on line 156 under the related work section)
>
> The reviewer proposed high-temperature sampling strategy can also be incorporated into our framework, with the objective reformulated as maximizing or minimizing the reward over the top $k$ candidates. This is a specialized scenarion of our framework. Notably, high temperature sampling when restricted to the top $k$ candidates ***still produces responses that lie within the same underlying distribution as our method***. This is because our decoding is performed only over the top $k$ tokens (with 𝑘 =5 in our experiments), rather than over the full entire vocabulary (eg. 128K tokens) of the LLM.
>
> We provide sample code for failure mode generation as evidence for the viability of our approach:
> Code: https://anonymous.4open.science/r/sample_code/README.md
>
> Thank you again for your time and effort in reviewing our paper! Please let us know if the above explanations do not address your concerns. In case these rebuttal addresses the concers of the reviewer we kindly request the reviwer to consider increasing their score.
>
> [1] Maxim Khanov, Jirayu Burapacheep, and Yixuan Li. Args: Alignment as reward-guided search,2024.
>
> [2] Yuancheng Xu, Udari Madhushani Sehwag, Alec Koppel, Sicheng Zhu, Bang An, Furong Huang, and Sumitra Ganesh. Genarm: Reward guided generation with autoregressive reward model for test-time alignment, 2025
>
> [3] Sidharth Mudgal, Jong Lee, Harish Ganapathy, YaGuang Li, Tao Wang, Yanping Huang, Zhifeng Chen, Heng-Tze Cheng, Michael Collins, Trevor Strohman, Jilin Chen, Alex Beutel, and Ahmad Beirami. Controlled decoding from language models, 2024.

---

> > ### Comment · Reviewer_7WHV · 2025-11-26
> >
> > Thank you for the responses to my question.
> > 1. I still maintain that the novelty is limited because of some similarities with the cited work. I disagree that 'high-scoring non-preferred responses' can't be generated by AdvRM, the framework allows for it as it tries to identify prompts where two reward models differ.
> > 2. I disagree that 'where only the preference data is available' is a more realistic setting.
> > There are similarities to the approach and the authors should atleast highlight that in the paper imo and possibly include it as a baseline. A learned approach can identify possibly more diverser prompts.
> >
> > I maintain my scores.

---

### Official Review · Reviewer_MxHx · 2025-11-02

**Soundness:** 2
**Presentation:** 2
**Contribution:** 2
**Rating:** 4
**Confidence:** 2

**Summary:**

This paper proposes REFORM, a novel framework for improving the robustness of reward models (RMs) used in aligning large language models (LLMs). The key idea is to let the RM act as its own adversary to discover and correct its failure modes. Instead of relying on external attribute priors or large human relabeling efforts, REFORM uses reward-guided controlled decoding to generate class-consistent but reward-inconsistent samples — responses that belong to the same preference class but are mis-scored by the RM. The method involves two stages:  failure discovery and targeted correction. Empirically, REFORM demonstrates consistent improvements in robustness on Anthropic Helpful–Harmless  and PKU Beavertails datasets using Mistral-7B and Qwen-14B reward models, without sacrificing in-distribution performance or downstream alignment quality under Best-of-N, PPO,  and DPO  training.

**Strengths:**

1. The paper offers a creative approach to reward model self-improvement. Its key innovation lies in using *the reward model itself* as a guide to generate adversarial counterexamples, circumventing the need for external attribute priors or large LLM queries.
2. The empirical performance shows the potential of the proposed method.

**Weaknesses:**

1. While the empirical findings are promising, the paper lacks deeper theoretical grounding on *why* self-adversarial discovery effectively generalizes across preference distributions. The surrogate objective (Eq. 6) and proxy use of partial reward evaluation are empirically justified but theoretically heuristic. A discussion on convergence guarantees or connections to robustness theory (e.g., adversarial training or influence functions) would strengthen the framework.

2. Despite that the high-level idea and the methodology proposed by the paper looks quite interesting, the introduction of detailed concepts and corresponding implementations are not satisfactory, please see my questions below. This casts concerns to the actual soundness of the method. Overall, I believe the question studies by the paper is very critical -- how to discover the failure mode or misspecified examples of a reward model without relying on prior attributes or distillating a stronger teacher but only using the reward model itself -- but the quality of presentation needs further improvement. I am willing to increase my rating if the concenrs are well addressed in the response.

**Questions:**

1. The introduction of the failure mode in Section 3.1 seems a little bit confusing to me: On the one hand, the reward model is assumed to be learned via the Bradley & Terry model (Eq (1)), a model which only specifies a relative order for a given response pair, i.e., given $y_1$ and $y_2$, either $y_1$ is preferred or $y_2$ is preferred. This is by the fact that the BT model only depends on the reward difference. However, later the authors introduce another concept "the preference class is preserved", where an explicit preferred or non-preferred label is assigned to a response. How is that defined? If a variant response pair only satisfies $y_+'$ is preferred than $y_-'$, it's ture that the BT model can not guarantee the relative order of $(y_+', y_-)$ and $(y_+, y_-')$. Is the concept of "class consistency" more like an extra constaint to the BT model?
2. What does "a policy aligned with the preference distribution" mean here? Does it mean a policy that has been fine-tuned on the preference dataset that is used to generate the failure mode?
3. I believe one of the core parts of the entire methodology proposed by the paper (see e.g. Figure 2) is that how to find the adversarially labeled responses that remain in the same "preference class" purely by the reward model itself. That is, how to ensure that a variant response searched is still in the same "preference class". But the algorithm part of the paper (Section 3) discusses nothing on this essential issue. How to guarantee that the searched response, e.g. through Eq (6), is going to be a preferred response? Also, how is the TopK hyperparameter choosen in practice? How is the  partial response $y_{<i}$ constructed?

---

> ### Author Response · Authors · 2025-11-15
> **Author rebuttal to Reviewer MxHx**
>
> * **Q1 Class consistency**
>
>
> Consider the scenario of generation a positive response $y_{+}^{'}$ that results in a lower reward
>
> Let's assume these exist as prompt $p$ and a pair of responses $y_{+}, y_{-}$. These response are assigned to a certain class $+$ or $-$ based on certain properties (in the case of harmfulness weather they relatively adhere to safety axioms or not). Given such a ranking for a given prompt the only baseline we have towards what should not belong to $-$ class is the response $y_{-}$. Thus a perfectly modelled reward model should rate any response $y_{+}^{'}$ that is similar to $y_{+}$ higher than $y{-}$. If it doesn't then we consider that the reward model have failed. This is in line with past works of [1]. A response  $y_{+}^{'}$ can only be considered as a response belonging to class $+$ if it would have ben rated as a $+$ class by the annotator. Thus the challenge becomes that of finding these responses $y_{+}^{'}$s that align with the annonators preferences and causes the model to rate response  $y_{+}^{'}$  lower than that of $y_{-}$.
>
> How we find these responses is what brings us to the generational part of the REFORM framework.
>
> A policy that is aligned in the preference distribution (in our case a policy that has be trained or test time aligned the preference distribution) will have a higher likelihood of generating tokens that are aligned with the preference distribution. Thus the framework we propose frames the problem as a constraint optimizion where we use the reward as a guidance in generating the sequence of tokens for which reward model will fail. Where
>
> **Constraint**: The space of tokens that are aligned with the preference distribution. Given the policy is trained to generate such tokens it is an appropirate assumtion that the top k tokens in this space will likely belong to the $+$ class.
>
> **Optimzation**: We want to find tokens that results in a lower reward. Thus to this end aim to minimize the reward and use it as a signal in guiding the generation from such an aligned policy.
>
> Similarly for the case of generating $y_{-}^{'}$ we frame the problem as a reward maximization with a misaligned policy (Equation 7)
>
> ---
>
> * **Q2 Definition on policy aligned with the preference distribution**
>
> It doesn't necessarily needs to be a policy finetuined on the preference distribution it can be also test time aligned with the imperfect reward.
>
> The goal of the alinged (or misaligned policy in the case of generating non preferred varaints) is to act as a tool in accessing the space of responses that are algined (or misaligned) with the perefrence distribution. In the presence of such a space the objective then becomes that of finding a response which induces a misspecification in the reward model.
>
> ***Why do we need a policy oriented approach for this generation***: RLHF captures human preference which is not often times defined in terms of axioms. In this paper we choose a harmlessness dataset as the base only because we want to evalute the framework's ability to generate reponses within the preference distribtuion as safety can be defined by axioms (more detail of the axioms on Appendix Code). But in general the preference distribution definition is open ended. Thus we popose to use a policy aligned/ misalgined on the preference distribution to find these samples. We do indeed show that in Figure 4 based on Gemini based LLM as a judge evaluation that these generated prompts are class consistent. In a real life setting one could additionally subject these prompts though another round of human feedback collection inorder to verify the class consitency of the generated samples.
>
> ---
>
> * **Q3 How to ensure the response stays in the preferec class**
>
> As mentioned above the preference distribtuion in general is not quantifable thus we need a proxy to generate likely samples in this space. This is why we use an algined/ misaligned policy in this ditribution as it likely to generate responses that are within the distribution. Emphrically we show that this class consistency is indeed maintained via LLM as a judge (using Gemini 2.5) evaluation in Figure 4 under class appropriateness.
>
> ---
>
> Thank you again for your time and effort in reviewing our paper! Please let us know if the above explanations do not address your concerns. We are happy to answer any further questions. If clarifications resolve the reviewer’s questions, we kindly request the reviewer to reconsider their score.
>
>
>
> [1] Junqi Jiang, Tom Bewley, Saumitra Mishra, Freddy Lecue, and Manuela Veloso. Interpreting language
> reward models via contrastive explanations, ICLR 2025

---

### Official Review · Reviewer_84Z8 · 2025-11-04

**Soundness:** 3
**Presentation:** 3
**Contribution:** 4
**Rating:** 8
**Confidence:** 2

**Summary:**

The paper proposes REFORM, a self-improving framework for reward models (RMs) trained from human preferences. The main motivation is that RMs, used pervasively in RLHF pipelines and test-time alignment, exhibit brittleness under distributional shift or targeted perturbations. Existing failure-discovery techniques typically require explicit knowledge of preference attributes (e.g., length, tone, toxicity) and thus do not generalize across domains or models.

REFORM introduces two key components:
	1.	Reward-guided failure discovery.  Instead of relying on external attributes, the method uses the reward model itself to generate class-consistent but reward-inconsistent samples—responses that belong to the same preference class (e.g., harmless, helpful) but are scored incorrectly. Controlled decoding guides the generation process by penalizing the reward score while keeping the response fluent and semantically aligned with the class. Formally, at each token, REFORM modifies the decoding objective
\min_{y_i \in \text{Top-K}} \ r_\theta(x,[y_{<i},y_i]) - \alpha \log \pi_D(y_i|x,y_{<i}),
steering toward low-reward variants for “preferred’’ responses (and symmetrically high-reward variants for “non-preferred’’ ones). This produces adversarial but meaningful failures.
	2.	Influence-targeted correction.  After generating these failures, the method fine-tunes the RM on a small augmented dataset built from the top 5% of “influential’’ training pairs—those with the lowest Bradley–Terry loss. The model is retrained (or fine-tuned) on these augmented pairs, thereby correcting the discovered mis-specifications without requiring broad relabeling.

Empirical evaluation uses Anthropic Helpful–Harmless and PKU Beavertails datasets with Mistral-7B and Qwen-14B reward models. REFORM is shown to:
	•	generate coherent and semantically valid failure examples without external attribute priors;
	•	improve robustness under four types of adversarial perturbations (verbosity, capitalization, repetition, misspelling);
	•	maintain in-distribution accuracy and downstream alignment quality (measured via BoN, PPO, and DPO policies).

The results suggest a 35–45% average gain in robustness with minimal degradation in reward quality.

**Strengths:**

1.	Conceptually elegant and well-motivated.  Turning the reward model into its own adversary is an appealing and general idea. The paper formalizes a clear operational notion of RM “failure’’ and exploits it for targeted data augmentation.
	2.	Technically sound within the RM framework.  The controlled-decoding formulation is simple yet effective. By combining reward minimization with language-model likelihood regularization, the authors achieve fluency while probing the RM’s vulnerabilities. The influence-based sampling for correction is principled and computationally light.
	3.	Empirical rigor.  Evaluation covers two standard safety datasets, two model families, and three alignment pipelines (BoN, PPO, DPO). Metrics include both in-distribution reward accuracy and out-of-distribution robustness. Figures 5–9 provide convincing evidence that REFORM improves robustness while preserving quality.
	4.	Practical impact.  The method requires no new labeling, no access to external attribute priors, and only a small fine-tuning budget. It could realistically be integrated into existing RLHF workflows.

**Weaknesses:**

1.	Limited theoretical analysis.  The paper lacks a quantitative characterization of the discovered failures or convergence guarantees for the fine-tuning step. One might ask whether the adversarial search reliably covers diverse failure modes or simply re-weights local perturbations around a few high-influence examples.
	2.	Restricted evaluation scope.  While robustness to perturbations is measured, the study focuses on specific lexical attacks. It would be valuable to see stress tests on semantic or reasoning-level shifts (e.g., long-context instructions, domain transfer).
	3.	Potential overfitting to adversarial decoding artifacts.  Because the fine-tuning samples are generated by the same reward model, there is a risk of reinforcing spurious patterns from the model’s own biases rather than true failures. A small discussion of safeguards (e.g., cross-model validation) would strengthen confidence.
	4.	Clarity and presentation.  The exposition is generally clear but occasionally dense; the decoding objectives (Eqs. 5–7) could benefit from a concise mathematical summary, and the distinction between class-consistency and semantic fidelity deserves clearer operationalization.

**Questions:**

1.	How sensitive is REFORM to the hyperparameter \alpha controlling fluency vs. reward deviation? Does a larger \alpha lead to trivial near-policy samples, while a smaller one yields incoherent outputs?
	2.	Since the failure discovery uses the same RM, how do you ensure diversity in the generated failures—could the method get trapped in a narrow mode of failure?
	3.	Can REFORM be iterated multiple times (self-play style) to progressively improve robustness? If so, does it converge or oscillate?
	4.	How would the framework handle non-text modalities or structured outputs where “class consistency’’ is harder to define?
	5.	In downstream alignment (PPO/DPO), does the observed robustness transfer when policies are further fine-tuned with new data, or is it specific to the evaluation distribution?

---

> ### Author Response · Authors · 2025-11-15
> **Author rebuttal to Reviewer 84Z8 (1/n)**
>
> * **Theoretical analysis** - The coverage and the likelihood of the policy in failure mode generation is proportional to the likelihood of a policy generating perference aligned responses. This likelihood depends on multiple factors that are difficult to quantify, as it can vary based on the knowledge and capabilities of the underlying policy LLM.
>
> ---
>
> * **Evaluation scope** - We only focused on the aspect of the reward model's vulnerabitlites in the RLHF framework and it's improvement. Our framework is not designed to capture all possible failure modes, as doing so would be intractable. Instead, we aim to demonstrate that, given a preference distribution, ***some of the reward model’s failure modes can be identified and corrected in a self-contained manner***.This is the reason why we did not consider vastly out of context test sets as the training of the reward model was done only on a limited preference dataset as opposed to a single large foundational dataset. Rather we wanted to showcase the applicability of the method accross multiple datasets and models in different scales.
>
> ---
>
> * **Potential overfitting to adversarial decoding artifacts** - To ensure that the reward model does not overfit to adversarial artifacts, we demonstrate that it maintains performance on downstream alignment tasks, including coherence and other quality metrics. If the reward model were overfitted to such artifacts, it would severely degrade performance in PPO-based alignment, as the policy would be guided to exploit these artifacts during training, potentially collapsing into artifact-driven generation for reward maximization (artifact based reward hacking).
>
> ---
>
> * **Clarity and presentation** - Thank your for the suggestion. We will modify the manuscript to incoporate the changes.
>
> ---
> * **Hyperparameter sensitivity** - In the case of a higher $\alpha$, the objective collpases into a trvial top k based sampling framework as now the objective is to merely maximize the probability of the LLM in the top k space with some randomness. In both the cases of larger top k and smaller alpha we saw the generations became incoherent as a conflicting reward minimization takes over the objective. For our experiments we maintained a value of k=5 and $\alpha$ = 1. For further technical carification we are adding an anonymized version of the implementation here - https://anonymous.4open.science/r/sample_code/README.md
>
> ---
>
> * **Narrow mode of failure** - While purely reward-model-guided generation carries the risk of collapsing into a narrow failure-mode distribution, our failure-mode generation is guided by both the pre-existing knowledge of the generative LLM and the reward model. The generalization of an aligned policy over a given preference distribution depends on both the knowledge embedded in the base LLM and the preference data itself. Consequently, selecting a stronger base model for generation can improve the diversity of collected failure samples.In our experiments, we made no explicit assumptions about the nature of reward failure modes. However, in scenarios where stronger assumptions exist, one could pair an LLM prompted with attribute guidance with REFROM to generate richer and more interpretable reward failure modes
>
> ---

---

> > ### Author Response · Authors · 2025-11-15
> > **Author rebuttal to Reviewer 84Z8 (2/n)**
> >
> > * **Scenarios where class consistency is not defined and other modalities** - We chose the harmlessness setting primarily for ease of evaluation, as existing safety axioms allow us to verify that our method preserves class consistency. However, in most real vworld scenarios, class definitions are open ended, making it impossible to explicitly define the underlying preference distribution. This motivates our proposal for ***data-guided generation using an aligned/misaligned policy***. If a policy is aligned in a distribution it is likely to generate samples that are consistant with the distribution ( we show that in Figure 4 that this is indeed the case). We argue that, in the absence of a predefined preference distribution, a data-centric approach is the ***most practical and viable strategy***. In real-world applications, one could further collect an additional round of human preferences over the generated samples to ensure class consistency. Furthermore, the framework is hypothetically expandable to other modalities as long as a next-token autoregressive generation framework exists (e.g., next-token image synthesis), the same principles can be applied.
> >
> > ----
> >
> > * **Iterative sample collection** - While iterative reward improvement is theoretically possible with our setup, it presents an interesting direction for future analysis due to the potential risk of model collapse. This risk arises from two factors:
> >     * Generated samples may become overfitted to existing reward vulnerabilities.
> >     * Reward models are being sequentially aligned on synthetic data, which may also lead to a model collapse.
> >
> > ---
> >
> > * **Downstream alginment** - In this work, we limited our scope to the robustness of the reward model, and we only performed downstream alignment to demonstrate that the reward model’s capabilities are preserved without degradation. Since our ***data augmentation was applied to the responses rather than the prompts***, we did not conduct robustness tests on the aligned policy with perturbed prompts.However, as shown in*** Appendix E.2 and F.8***, we qualitatively observed that the reward model ***avoids spurious correlations*** when rating responses and ***slightly improves the performance*** of the downstream policy.
> >
> > ----
> >
> >
> > We sincerely thank the reviewer for their time and effort in reviewing our paper. Please let us know if the above explanations do not address your concerns. We are happy to answer any further questions. If you find our responses satisfactory, we kindly request that you consider increasing your score or the confidence.

---

### Note · Authors · 2026-01-03

I have read and agree with the venue's withdrawal policy on behalf of myself and my co-authors.